# Learning from Examples and Self-Exploration: A New Paradigm for Dynamic Fusion

## Abstract

Alignment of Large Language Models with human preferences is dominated by two paradigms: Supervised Fine-Tuning (SFT) and Reinforcement Learning (RL), exemplified by methods like Group Relative Policy Optimization. Yet, they face a trade-off challenge: SFT excels at incorporating external knowledge but often fails to foster deep comprehension, whereas RL can internalize knowledge but struggles to expand the model knowledge frontier. To resolve this, we propose **LESE** (**L**earning from **E**xamples and **S**elf-**E**xploration), a framework that dynamically interpolates between SFT and RL. LESE introduces an instance-adaptive mechanism that assesses a model real-time task proficiency and exploration diversity, thereby allocating a dynamic weight between SFT and RL for each training instance. This adaptive methodology addresses the limitations of static strategies by adjusting the balance between SFT and RL at the instance level. Empirically, it improves performance on mathematical benchmarks and enhances training stability, while maintaining consistency with human-preferred outputs.

## 1 Introduction

The advent of Large Language Models (LLMs) has marked a significant advancement in the field of artificial intelligence (Achiam et al., 2023; Zhao et al., 2023). Notably, the recent emergence of Large Reasoning Models (LRMs), has demonstrated immense potential for tasks requiring complex symbolic manipulation and rigorous logic, such as mathematical reasoning (Guo et al., 2025; Jaech et al., 2024; Team et al., 2025). However, a core challenge lies in translating this potential into stable, reliable, and generalizable reasoning capabilities (Yan et al., 2024). Current models often struggle with a fundamental dilemma between "shallow imitation" and "inefficient exploration". On one hand, they may resort to rote memorization of exemplars, which prevents them from effectively addressing novel problems. On the other hand, their attempts at autonomous exploration frequently cause them to deviate from valid logical pathways, leading to a breakdown of the reasoning process.

To enhance the reasoning capabilities of large models, researchers primarily employ two technical paradigms: Supervised Fine-Tuning (SFT) (Shengyu et al., 2023) and Reinforcement Learning with Verifiable Rewards (RLVR) (Zhang et al., 2025a). SFT is efficient for instilling knowledge by training the model to imitate high-quality solution demonstrations. However, this approach suffers from an inherent "imitation trap". In mathematical reasoning, this manifests as the model memorizing solution templates rather than internalizing the underlying mathematical principles. Consequently, performance falters on problems that demand creative thinking or complex, multi-step inference. Conversely, RLVR aims to reinforce a model reasoning abilities through autonomous exploration of a reward landscape, which theoretically allows it to surpass the performance ceiling imposed by SFT. Nevertheless, this approach introduces significant "exploration risks" and instabilities. For instance, the sparse rewards endemic to complex proof-generation can lead to vanishing gradients, while the model may also engage in reward hacking—exploiting the reward function to produce logically incoherent pseudo-proofs.

Given the complementary nature of these two strategies, the research community has begun to explore hybrid training schemes. Early "SFT-then-RL" pipelines, such as InstructGPT (Ouyang et al., 2022), treated imitation and exploration as two distinct, sequential stages. This evolution

Figure 1: Conceptual Comparison of Training Paradigms: From Separation to Adaptive Fusion.

of training paradigms, from simple sequential application to more integrated hybrid approaches, is conceptually illustrated in Figure 1. Although more sophisticated static fusion methods subsequently emerged—including fixed SFT-RL sequences and static weighting techniques (e.g., RPO, KDRL)—these approaches remain fundamentally static and coarse-grained (Liu et al., 2024; Xu et al., 2025). They are incapable of adaptively adjusting to the model evolving reasoning capabilities or the intrinsic difficulty of different mathematical problems. Consequently, they struggle to strike an optimal balance between training stability and exploration efficiency.

To address this core dilemma, we introduce a unified alignment principle designed to foster deep reasoning. Our approach integrates the imitation-based learning of SFT with the exploration-based learning of RLVR within a single, dynamic framework, as shown in Figure 1. This framework, which we call LESE (Learning from Examples and Self-Exploration), applies a real-time, fine-grained weighting to each training instance. The weighting is determined by two key diagnostic metrics: the model "Task Mastery" and its "Exploration Diversity" for that specific problem. To this end, we design and implement LESE (Learning from Examples and Self-Exploration), an instance-adaptive alignment algorithm. We conduct a comprehensive evaluation of LESE on four competition-level mathematics benchmarks (Hendrycks et al., 2021b; Li et al., 2024) and the out-of-distribution (OOD) GPQA-Diamond (Rein et al., 2024) benchmark to validate its effectiveness in enhancing both core reasoning ability and generalization performance.

Our main contributions can be summarized as follows.

1. **A unified dynamic alignment paradigm.** We propose LESE, an adaptive framework that unifies SFT and RL via instance-level interpolation. The weight is determined by two diagnostic metrics—Task Mastery and Exploration Diversity—allowing dynamic adjustment of learning strategies per sample for both stability and efficiency.

2. **A fine-grained preference reward design.** We propose a multi-component reward function that provides dense feedback by decomposing human preferences into verifiable sub-signals. This design is aimed to accelerate convergence, improve training stability, and ensure long-term preference alignment.

3. **Extensive empirical validation.** We evaluate LESE on multiple challenging mathematical reasoning tasks and the out-of-distribution GPQA benchmark. Experiments show that LESE not only achieves state-of-the-art (SOTA) performance, but its effectiveness also persists across different model scales, demonstrating its robustness and scalability.

## 2 RELATED WORKS

**Supervised Fine-Tuning (SFT).** Supervised Fine-Tuning (SFT) adapts a pretrained LLM for downstream tasks by minimizing the cross-entropy loss between the model's predictions and the ground-truth sequences within a corpus of expert demonstrations (Hendrycks et al., 2021a). Formally, the single-sample SFT loss is defined as:

$$\mathcal{L}_{SFT}(\theta) = -\frac{1}{T} \sum_{t=1}^{T} \log \pi_\theta(y_t|x, y_{<t}) \tag{1}$$

where $(x, y)$ denotes input-output pairs from example datasets. Its efficiency and low cost make it attractive, but SFT primarily encourages rote imitation, leading to the "imitation trap" (Chen et al., 2025), where models struggle to generalize beyond training distributions or surpass the quality ceiling of exemplars (Chu et al., 2025).

**Reinforcement Learning (RL).** To overcome these limitations, reinforcement learning (RL) offers a powerful paradigm. Instead of merely mimicking a static dataset, RL enables the model to actively explore the solution space and learn from feedback signals (rewards), potentially generating outputs that surpass the quality of the original demonstrations. While various RL algorithms exist, such as PPO (Schulman et al., 2017), DPO (Rafailov et al., 2024) and Group Relative Policy Optimization (GRPO) (Shao et al., 2024) has emerged as a particularly efficient approach. GRPO bypasses the need for a separate value model by sampling multiple candidate solutions for a prompt and using their relative reward scores to directly estimate the advantage for the policy update. Our formulation omits the KL divergence term to avoid constraining the performance of the policy LLM and adopts the decoupled upper and lower clipping strategy from DAPO (Yu et al., 2025). Formally, the GRPO loss is defined as:

$$\mathcal{L}_{\text{GRPO}}(\theta) = -\frac{1}{\sum_{i=1}^{N} |o_i|} \sum_{i=1}^{N} \sum_{t=1}^{|o_i|} \min \left[ r_{i,t}(\theta) A_i, \text{clip}\left(r_{i,t}(\theta), 1 - \epsilon_{low}, 1 + \epsilon_{high}\right) A_i \right] \quad (2)$$

where $r_{i,t}(\theta) = \frac{\pi_\theta(o_{i,t}|q,o_{i,<t})}{\pi_{\theta_{old}}(o_{i,t}|q,o_{i,<t})}, A_i = \frac{R(o_i)-\text{mean}(R(o_i))}{\text{std}(R(o_i))}$. This technique is effective in Reinforcement Learning with Verifiable Rewards (RLVR), enhancing complex reasoning under resource constraints. However, RL methods still face key challenges: reward hacking (exploiting reward function flaws), sparse/noisy rewards, and over-optimization—all of which narrow output distribution, reduce diversity, and hinder genuine reasoning improvement (Pan et al., 2022; Dietterich, 2000). Another critical limitation is RL's difficulty in expanding knowledge boundaries: it typically learns within predefined task scopes and data distributions, failing to generalize to scenarios beyond initial training (Yue et al., 2025; He et al., 2025).

**Static-Hybrid Fusion.** In order to combine the respective merits of SFT and RL to achieve complementarity, hybrid strategies were developed. Early pipelines such as InstructGPT (Ouyang et al., 2022) applied RL after SFT, while recent works designed static or coarse-grained fusion schemes. For example, ReLIFT (Ma et al., 2025) interleaves SFT with RL on difficult cases, RPO (Liu et al., 2024) interprets SFT loss as an implicit regularizer against over-optimization , Chord-$\mu$ (Zhang et al., 2025b) adjusts the trade-off via a global decay factor, and LUFFY (Yan et al., 2025) combines offline and online trajectories. Large-scale efforts such as DeepSeek-R1 (Guo et al., 2025) further demonstrate the potential of combining the two paradigms. However, these methods remain fundamentally static, unable to adapt to the evolving competence of the model or the difficulty of individual samples, and are thus prone to overfitting or training instability.

## 3    LESE: A DYNAMIC FUSION OF LEARNING FROM EXAMPLES AND SELF-EXPLORATION

Recent theoretical results have revealed structural similarities between the gradient formulations of Supervised Fine-Tuning (SFT) and Reinforcement Learning (RL), motivating a more unified optimization perspective (Wu et al., 2025; Lv et al., 2025). Building on this insight, and to overcome the "imitation trap" of SFT and the "exploration risk" of RL, we introduce LESE (Learning from Examples and Self-Exploration), an instance-level adaptive alignment framework. An overview of the LESE training framework is depicted in Figure 2.

The core of LESE is a dynamic fusion mechanism that replaces static training schedules with a continuous, data-driven adjustment between SFT and RL. By diagnosing the model task mastery and exploration diversity in real time for each training sample, LESE dynamically balances the two paradigms. This fine-grained adaptive mechanism distinguishes LESE from prior static hybrids and establishes a more robust paradigm for reasoning-oriented alignment.

For each training instance, the interpolation coefficient $\alpha \in [0, 1]$ determines the relative contributions of the SFT and RL objectives:

$$L_{\text{LESE}} = (1 - \alpha) \cdot L_{\text{SFT}} + \alpha \cdot L_{\text{GRPO}} \quad (3)$$

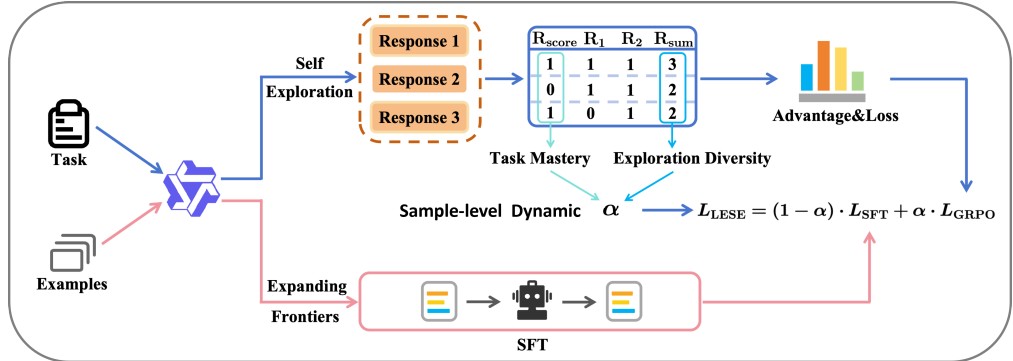

Figure 2: **Overview of LESE Training Framework.** Unifies SFT and RL by introducing an instance-level adaptive coefficient $\alpha$, which dynamically adjusts the contribution of each objective for every training sample.

The coefficient $\alpha$ is not fixed but is dynamically adjusted according to the model's task mastery and exploration diversity, two diagnostic metrics designed to capture complementary aspects of the learning state.

### 3.1 CORE DIAGNOSTIC METRICS

For each training instance (i.e., a single prompt), we generate G responses and evaluate their quality using a set of K reward components.

**Task Mastery ($M$):** This metric quantifies whether the model has acquired the minimum level of competence required for a given task. It is defined as the fraction of generated responses in a batch that exceed a threshold reward:

$$M = \frac{1}{G} \sum_{i=1}^{G} \mathbb{I}(R_{\text{core}}(o_i) \geq R_{\text{thres}})$$

(4)

where $R_{\text{core}}$ denotes the core reward obtained for response $o_i$. Low mastery ($M \to 0$) indicates that RL optimization may fail due to sparse or noisy rewards. High mastery suggests that the model has acquired the prerequisites to benefit from exploratory optimization.

**Exploration Diversity ($D$):** This metric captures the variability of reward outcomes across generated responses, computed as the standard deviation of the total rewards:

$$D = \text{StdDev}\left(\left\{\sum_{i=1}^{N} R_{j,i}\right\}_{j=1}^{G}\right)$$

(5)

High diversity implies that exploration is yielding a rich variety of outputs, providing informative gradients for RL. Conversely, low diversity ($D \to 0$) suggests that responses have become homogeneous, increasing the risk of overfitting to noise or spurious reward signals.

### 3.2 DESIGN OF LESE DYNAMIC INTERPOLATION FUNCTION

The interpolation coefficient $\alpha$ is determined by the diagnostic metrics—Task Mastery ($M$) and Exploration Diversity ($D$)—according to the following function:

$$\alpha = \begin{cases} \min\left(1, l(e^M - 1)\frac{D}{D_{\max}}\right) & \text{if } M > c \\ 0 & \text{if } M \leq c \end{cases}$$

(6)

where $c$ is the mastery threshold, $l$ is a scaling factor, and $D_{max}$ is the theoretical maximum standard deviation of in-group rewards. This design incorporates three core principles:

**Mastery as a prerequisite for exploration.** When task mastery falls below the threshold $c$, the model lacks sufficient capability to interpret reward signals effectively. In this regime, exploration through RL is unlikely to provide meaningful gradients due to sparse or noisy rewards. To avoid unstable updates, $\alpha$ is set to zero, and the optimization relies solely on SFT, which provides a stable supervised signal for consolidating fundamental knowledge.

**Non-linear amplification of mastery.** Once mastery exceeds the threshold, the exponential term $e^M - 1$ gradually increases the weight assigned to RL. This reflects the intuition that the marginal utility of exploration grows non-linearly with task competence: as the model becomes more proficient, the benefits of exploring alternative strategies or solutions increase at a faster rate. The exponential scaling ensures a smooth yet decisive shift toward exploration as competence improves.

**Diversity as a stability regulator.** The factor $D/D_{\max}$ evaluates the variability of reward signals across generated responses. High diversity indicates that exploration yields informative gradients, while low diversity suggests that responses have become homogeneous, making RL updates unreliable. By proportionally scaling down $\alpha$ when diversity is low, this mechanism reduces the risk of policy collapse and prevents overfitting to spurious reward signals.

In summary, the interpolation function balances stability and adaptivity: SFT dominates when the model has not yet acquired sufficient competence or when exploration signals are unreliable, while RL becomes increasingly emphasized as mastery and exploration diversity rise. This formulation grounds the adaptive mechanism in measurable properties of the learning process, ensuring both robustness and efficiency.

### 3.3 Dynamic Pedagogical Strategy of LESE

LESE transforms the conventional static pipeline into an adaptive process that adjusts the balance between SFT and RL at the instance level. This adjustment produces a continuous trade-off with smooth transitions between two distinct regimes:

**Learning from External Examples ($\alpha \to 0$).** When task mastery is below the threshold ($M < c$) or its exploration diversity is negligible ($D \approx 0$), the update relies entirely on SFT. In this regime, the dense and reliable gradients from supervised learning strengthen the model fundamental capabilities and ensure stable optimization. This design mitigates the risks of gradient vanishing or spurious updates in RL, which typically occur when reward signals are sparse or noisy.

**Learning from Self-Exploration ($\alpha \to 1$).** When the model demonstrates sufficient competence ($M \gg c$) and exploration diversity is high ($D \gg 0$), the weight shifts toward RL. In this regime, optimization emphasizes policy refinement, encouraging exploratory reasoning that pushes the model beyond the ceiling of the original supervised data. This helps the model to overcome the limitations of SFT, such as shallow pattern imitation and overfitting, thereby extending its reasoning capability beyond examples.

In addition, LESE incorporates a mechanism for filtering saturated instances. When all responses in a batch achieve the maximum reward, the update is skipped. In such cases, RL produces zero advantage, and further SFT updates risk overfitting without providing additional benefit. This filtering, combined with the dynamic weighting mechanism, ensures that training signals remain both effective and efficient, preventing wasted computation and improving overall alignment stability.

While the dynamic fusion mechanism adaptively balances SFT and RL at the instance level, its effectiveness ultimately depends on the quality and granularity of the reward signal. We therefore turn to the design of preference rewards, which directly determine the stability and efficiency of policy optimization.

### 3.4 Fine-Grained Rewards Accelerate Convergence

A central challenge in RL-based alignment is the sparsity and noisiness of reward signals, particularly when targeting complex human preferences. If the preference criterion is overly strict or binary, the model may rarely generate outputs that merit a positive reward, leading to vanishing gradients and stalled convergence. To address this, we design a fine-grained reward strategy that decomposes the alignment objective into multiple verifiable sub-signals, such as tag presence, ordering, and format compliance. Each satisfied sub-condition contributes a partial reward, enabling incomplete

outputs to provide informative gradients. This increases feedback density, mitigates reward sparsity, and accelerates early-stage learning while remaining consistent with human-preferred outputs.

As illustrated in Figure 3, reward granularity critically shapes training dynamics. With fine-grained signals, both LESE and GRPO can make progress, but LESE converges rapidly and maintains stable preference rewards throughout training, whereas GRPO exhibits volatility and eventual degradation. When restricted to binary rewards, the contrast is sharper: *LESE-binary* eventually converges but requires a substantially longer training horizon due to the sparsity of signals, while GRPO fails to achieve meaningful reward at all.

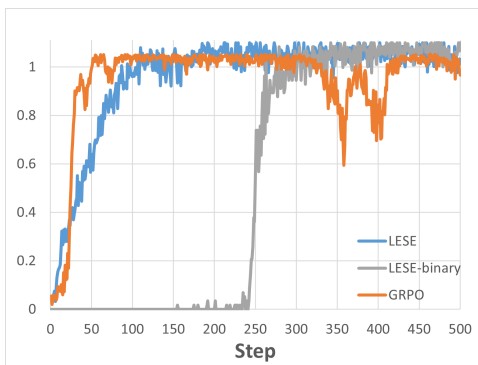

Figure 3: Training dynamics under fine-grained and binary format rewards. The detailed of $R_{\text{format}}$ will be provided in the appendix A.5

These findings highlight two advantages of our design. First, fine-grained signals are sufficient to guide effective policy optimization, accelerating convergence and improving calibration without requiring late-stage transitions to stricter criteria. Second, the adaptive fusion in LESE ensures robustness across different reward granularities, preventing collapse even when feedback is sparse—an ability that GRPO lacks.

## 4 EXPERIMENTS

### 4.1 EXPERIMENTAL SETUP

**Datasets.** Our experiments utilized a mixed curriculum learning dataset comprising 32,000 samples. This dataset is constructed by combining two sources: 30,000 challenging math problems curated from the OpenR1-Math-220k (Face, 2025) dataset and 2,000 relatively simple samples from the GSM8K (Cobbe et al., 2021) dataset. To simulate a progressively challenging learning process, the dataset is structured as a curriculum with a higher proportion of simpler samples in the initial training stages, gradually increasing the ratio of difficult samples as the model proficiency improves. Further details are provided in Appendix A.1.

**Training.** All experiments are conducted on the qwen2.5-math-1.5b (Yang et al., 2024) model. To enhance the model explorative capabilities and maintain consistency with baseline methods, we expand the RoPE theta from 10,000 to 40,000 and increase the context window size to 16,384. Across all training runs, we consistently use a context length of 8192, a batch size of 16, and a learning rate of $5 \times 10^{-6}$. For the RL-based methods, training is conducted for 500 steps, while SFT is trained for 3 epochs. During the RL phase, we set the sampling temperature to 1.0 and remove the KL-divergence term. The detail and specifics of our reward function are detailed in Appendix A.2.

**Baselines.** We conduct a comprehensive comparison of LESE against four representative baseline methods. These include two RL-centric approaches: (1) Simple-RL (Zeng et al., 2025) : train models using a rule-based reward function, and (2) LUFFY (Yan et al., 2025), combines offline and online trajectories. We also compare against two SFT-RL hybrid methods: (3) ReLIFT (Ma et al., 2025), interleaved online fine-tuning for hardest questions, and (4) Chord-$\mu$ (Zhang et al., 2025b): balances SFT and RL via a global decay factor.

**Evaluation.** Model performance is evaluated on four mainstream mathematical reasoning benchmarks: AIME 2024, AIME 2025, AMC (Li et al., 2024), and MATH-500 (Hendrycks et al., 2021b). Furthermore, we assess the generalization capabilities of the model on an out-of-distribution benchmark, GPQA-Diamond (Rein et al., 2024). For AIME and AMC, which have smaller test sets, we report *avg@32* scores. For the other datasets, we use *pass@1* as the evaluation metric. All evaluations are conducted with a temperature of 0.6 and a maximum generation length of 8192.

Table 1: The table compares LESE with base models, single-paradigm approaches, and hybrid strategies on five benchmarks. **Bold** and underline denote the **best** and second-best results, respectively.

| Model | AIME24 | AIME25 | MATH-500 | AMC | GPQA | Avg. |
|-------|--------|--------|----------|-----|------|------|
| Qwen-Math-1.5B | 3.6 | 2.2 | 45.2 | 29.8 | 13.6 | 18.9 |
| Qwen-Math-Instruct | 6.8 | 8.2 | 76.2 | 44.3 | 25.3 | 32.2 |
| Deepseek-Distill-Qwen | 9.8 | **13.4** | 75.6 | 42.5 | **30.3** | 34.3 |
| SimpleRL-Zero | 4.2 | 5.6 | 59.0 | 12.5 | 17.5 | 19.8 |
| ReLIFT | 7.6 | 8.4 | 74.8 | 52.3 | 21.2 | 32.9 |
| LUFFY | 8.5 | 11.1 | 76.5 | 50.0 | 25.2 | 34.2 |
| CHORD-$\mu$ | 6.6 | 6.8 | 76.0 | 48.9 | 25.8 | 32.8 |
| SFT | 7.6 | 12.0 | 65.4 | 40.2 | 9.6 | 27.0 |
| RL | 6.4 | 7.2 | 71.6 | 36.3 | 22.2 | 28.7 |
| SFT then RL | 7.8 | 9.0 | 73.7 | 46.4 | 23.2 | 32.0 |
| LESE | **10.0** | 13.1 | **79.4** | **55.5** | 28.9 | **37.4** |

## 4.2 MAIN RESULT

**Superior Reasoning Performance.** As detailed in Table 1, our proposed method, LESE, establishes a new state-of-the-art on the **Qwen2.5-Math-1.5B** model, achieving an average accuracy of 37.3% to surpass a comprehensive suite of baselines. Crucially, this superior performance on the main task is achieved while simultaneously adhering to stringent human preference rewards, highlighting that our approach enhances reasoning capabilities without sacrificing alignment.

The advantages of LESE are particularly notable when compared to single-paradigm approaches. It surpasses pure Supervised Fine-Tuning (SFT) and pure Reinforcement Learning (RL) by 10.3 and 8.6 percentage points in average accuracy, respectively. This outcome highlights the inherent limitations of static strategies: SFT is constrained by its "imitation trap", while RL is susceptible to "exploration risks" that can hinder performance.

Beyond outperforming single methods, LESE also improves upon coarse-grained hybrid approaches. It achieves a **4.5** point improvement over Chord-$\mu$, the strongest static fusion baseline, and further exceeds the off-policy guidanced LUFFY method by **2.9** points. These results provide strong evidence for the effectiveness of the adaptive mechanism of LESE. Unlike static pipelines—whether sequential(e.g., SFT then RL), interleaved(e.g., ReLIFT), or globally weighted (e.g., Chord-$\mu$)—LESE dynamically adjusts the weighting of SFT and RL at the instance level. This fine-grained adaptivity aligns training with the evolving learning state of the model, yielding more efficient knowledge acquisition, enhanced reasoning capability, and improved generalization.

**Scalability and Generalization.** To validate the scalability of LESE, we extended its application to the larger **Qwen2.5-Math-7B** model. Notably, our approach demonstrates superior sample efficiency and stability, surpassing the baseline's performance despite utilizing an even smaller batch size, a larger learning rate, and significantly fewer rollouts per prompt. Due to page limits, detailed results are deferred to Appendix A.3. These findings confirm that LESE is a highly scalable alignment strategy, delivering consistent performance gains over single-paradigm methods across different model scales.

**Analysis of Inference-Time Behavior.** To better understand the inference-time characteristics of each method, we analyzed the average response length for both correctly and incorrectly solved problems, with results presented in Table 2. This analysis reveals distinct strategic tendencies shaped by the different alignment approaches.

The results reveal LESE adaptive inference-time strategy. On problems the model solves correctly, LESE generates markedly concise responses, second only to the pure RL policy in brevity. This indicates that for tasks within its competence, LESE defaults to an efficient, direct reasoning path. Conversely, on problems it fails to solve, LESE produces substantially longer responses than all

Table 2: Average response lengths (in tokens) across benchmarks.The table reports mean response lengths for correct (T) and incorrect (F) answers under different alignment methods. These results allow us to examine reasoning efficiency (on correct solutions) and exploratory behavior (on incorrect attempts).

| Model | AIME24 | | AIME25 | | MATH-500 | | AMC | | GPQA | | Overall | |
|-------|------|------|------|------|------|------|------|------|------|------|------|------|
| | T | F | T | F | T | F | T | F | T | F | T | F |
| BASE | 3737 | 3945 | 4132 | 3929 | 742 | 4647 | 1989 | 2263 | 1132 | 2647 | 1701 | 3460 |
| LUFFY | 4135 | 4959 | 4053 | 3919 | 1422 | 3196 | 2328 | 3336 | 2452 | 2971 | 2278 | 4042 |
| ReLIFT | 4151 | 4869 | 3676 | 4842 | 1250 | 4143 | 2249 | 5827 | 1505 | 4624 | 2159 | 4932 |
| RL | 1675 | 2370 | 1965 | 2415 | 518 | 2445 | 1710 | 3835 | 849 | 1798 | 1258 | 2770 |
| SFT | 8192 | 8191 | 8192 | 8192 | 8181 | 8192 | 8176 | 8192 | 8192 | 8192 | 8181 | 8192 |
| LESE | 2651 | 6625 | 2713 | 4447 | 818 | 6046 | 1995 | 3741 | 2798 | 3278 | 1801 | 5035 |

other methods. This suggests that when confronted with a challenge, LESE does not give up but instead engages in more persistent and deep exploration by constructing extended lines of thought.

Taken together, these contrasting behaviors underscore the effectiveness of LESE dynamic trade-off mechanism. The model intelligently adapts its approach to the problem at hand—employing a concise style for solvable tasks while shifting to an exploratory mode for more difficult ones. This adaptability enables LESE to effectively balance reasoning efficiency with exploratory depth, thereby mitigating the risk of over-optimization (policy sharpening) seen in pure RL and overcoming the superficial imitation characteristic of pure SFT.

## 4.3 TRAINING DYNAMICS

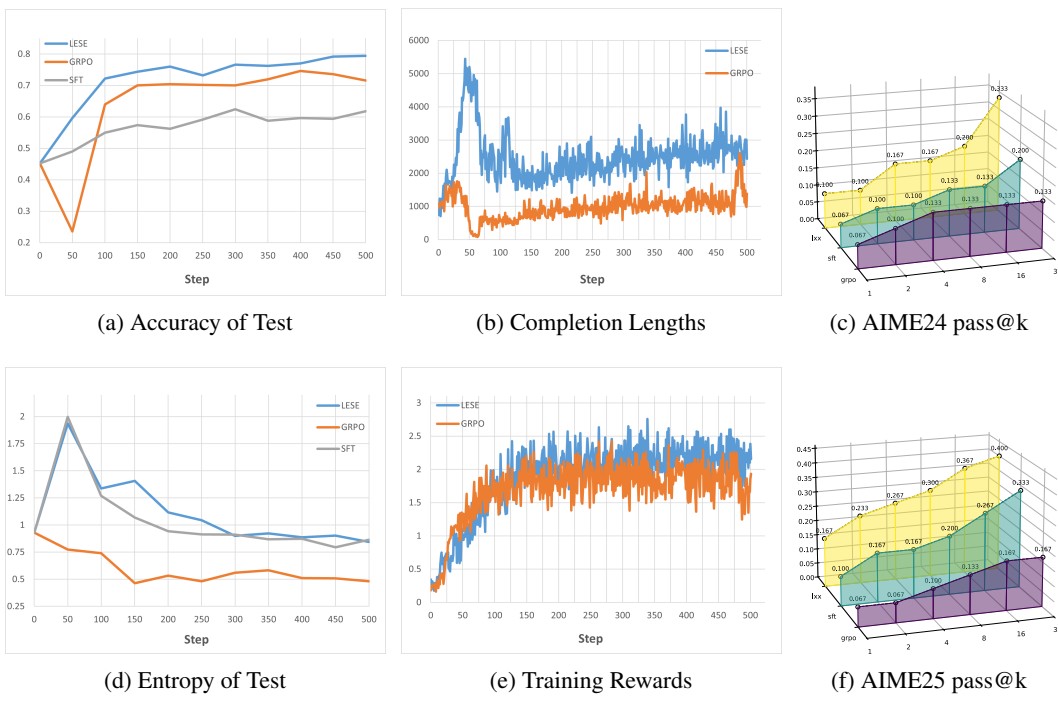

(a) Accuracy of Test  (b) Completion Lengths  (c) AIME24 pass@k

(d) Entropy of Test  (e) Training Rewards  (f) AIME25 pass@k

Figure 4: Training dynamics of different alignment strategies. (a,d) Test accuracy and generation entropy of SFT, RL, and LESE. (b,e) *pass@k* performance on the AIME benchmark. (b,e) Training reward and corresponding response length of LESE and RL.

To further validate the effectiveness of LESE, we analyze its training dynamics in comparison with pure SFT and RL. As shown in Figure 4a, LESE exhibits steadily improving test accuracy throughout

training, closely mirroring the stability of SFT. In contrast, the pure RL policy quickly plateaus due to reward hacking, where the optimization diverges from the intended task objective.

This divergence is also reflected in the generation behavior and exploration patterns. The RL policy distribution sharpens rapidly, favoring shorter responses that optimize secondary signals rather than core reasoning (Figures 4a, 4b). Consequently, while total rewards continue to increase (Figure 4e), task accuracy and response length decline sharply, accompanied by a collapse in policy entropy (Figure 4d).

In comparison, LESE maintains high policy entropy at a level comparable to SFT while simultaneously demonstrating greater adaptability. It first adapts to an extended context length and then dynamically adjusts the response length according to sample difficulty, leading to more stable optimization. More importantly, as shown in Figures 4c, 4f, LESE achieves superior *pass@1* performance, reflecting a deeper mastery of the underlying reasoning tasks. In addition, its *pass@k* results show consistent improvements with larger k, highlighting that LESE preserves exploratory diversity that is often lost in pure RL policies.

Overall, the training dynamics confirm that LESE mitigates the exploration risk inherent in RL while simultaneously escaping the imitation trap of SFT. The training dynamics indicate that LESE mitigates the instability observed in RL while avoiding the stagnation of pure SFT. By adjusting the interpolation coefficient based on learning signals, the method sustains policy entropy and prevents premature convergence.

## 4.4 ABLATION STUDY AND ANALYSIS

We conduct ablation studies to isolate the contributions of the adaptive components of LESE (Table 3). Non-adaptive variants—LESE-Static($\alpha = 0.5$) and LESE-Random ($\alpha \sim U(0, 1)$)—are unstable and prone to reward hacking, with formatting optimized at the expense of reasoning. Single-metric variants reveal complementary weaknesses: LESE-M (mastery only, we set $D = D_{max}$) prematurely triggers RL updates under low diversity, leading to over-sharpened policies, while LESE-D (diversity only , we set $(e^M - 1) = 1$) emphasizes exploration without sufficient competence, causing instability on in-domain tasks. Interestingly, LESE-D achieves the best out-of-distribution (OOD) generalization on GPQA-Diamond (Rein et al., 2024), suggesting unconditional exploration may be advantageous in unfamiliar domains.

Table 3: Ablation study of LESE and its variants across benchmarks. For AIME/AMC, we report both *avg@32* and *pass@32*; for MATH-500 and GPQA-Diamond, we report *pass@1*. The last column (*Format*) measures adherence to human-preferred formatting, where 0 indicates reward hacking or local optima. Best results are in **bold**.

| Model | AIME24 *avg@32/pass@32* | AIME25 *avg@32/pass@32* | AMC *avg@32/pass@32* | MATH-500 *pass@1* | GPQA *pass@1* | Format *Binary* |
|---|---|---|---|---|---|---|
| LESE | **10.0 / 33.3** | **13.1 / 40.0** | **55.5 / 92.5** | **79.4** | 28.9 | **95.4** |
| LESE-Random | 7.8 / 20.0 | 6.7 / 23.3 | 45.0 / 85.0 | 72.8 | 19.7 | 0 |
| LESE-Static | 4.6 / 20.0 | 6.3 / 26.7 | 37.5 / 85.0 | 71.0 | 20.2 | 0 |
| LESE-$M$ | 7.3 / 26.7 | 5.6 / 26.7 | 52.2 / 90.0 | 75.4 | 27.3 | 94.2 |
| LESE-$D$ | 8.3 / 30.0 | 9.7 / 36.7 | 46.3 / 90.0 | 76.2 | **30.8** | 95.0 |

The full LESE framework, guided jointly by mastery and diversity, consistently achieves the strongest performance and alignment. This confirms that both metrics are necessary: mastery grounds exploration in competence, while diversity prevents policy collapse. Together they enable a robust, efficient, and generalizable alignment mechanism.

## 5 CONCLUSION

We proposed LESE, a dynamic fusion of supervised fine-tuning and reinforcement learning guided by task mastery and exploration diversity. LESE achieves stable training, faster convergence, and superior reasoning performance across benchmarks. Our results establish LESE as a scalable paradigm for aligning reasoning models, with promise for broader domains beyond mathematics.

## REPRODUCIBILITY STATEMENT

We are committed to ensuring the reproducibility of this research. All experiments are based on publicly available models and datasets. We detail the dataset construction process in Appendix A.1, including the filtering, mixing, and curriculum learning setup for the source data (OpenR1-Math-220k (Face, 2025) and gsm8k (Cobbe et al., 2021)). Detailed experimental settings, including the model (Qwen2.5-math-1.5B and Qwen2.5-math-7B (Yang et al., 2024)) , training hyperparameters (e.g., learning rate, batch size) , and specific parameters for our proposed LESE method, are provided in Section 3.1, 3.2 and Appendix A.5. The evaluation benchmarks and metrics are also specified in Section 4.1. To facilitate reproducibility, we will make all our code publicly available at `https://anonymous.4open.science/r/LESE-6418`.

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

## A    APPENDIX

### A.1    DATASETS

We constructed a curriculum learning dataset with progressively increasing difficulty using the following sources:

- **Math-Filtered.** We began with the open-r1/OpenR1-Math-220k-default (Face, 2025) dataset, which was distilled from DeepSeek-R1 (Guo et al., 2025). To ensure data quality, we performed a rigorous preprocessing pipeline: (1) filtering out samples with hyperlinks, or mixed Chinese-English text; (2) removing entries that failed verification by Math-Verify (Face, 2024); and (3) discarding samples exceeding a context length of 8192. This process yielded approximately 30,000 high-quality samples.
- **GSM8K.** We utilized the classic openai/gsm8k (Cobbe et al., 2021) dataset, which consists of elementary school math word problems. We manually added the required labels and selected approximately 2,000 samples, which are considered relatively simple.

Mixed Curriculum Dataset: To simulate a learning process of increasing difficulty, we blended the Math-Filtered and GSM8K (Cobbe et al., 2021) datasets. The full dataset of 32,000 samples was divided into 320 groups. For each group of data, weights need to be dynamically assigned to the two datasets: the weight of the GSM8K dataset is denoted as $w_G$, and the weight of Math-Filtered is correspondingly set to $(1 - w_G)$; furthermore, for each group $w_G$) is calculated using the following formula:

$$w_G(i) = w_i \cdot \left( 1 - \left( \frac{i}{N_{groups} - 1} \right)^p \right) \tag{7}$$

This strategy adapts to the model rapidly growing capabilities in the early stages of training. We set the initial weight $w_i$ to 0.65 and the exponent p to 0.11. We used all the 32k data to construct the dataset.

## A.2 TRAINING

Our primary experiments were conducted on the **Qwen2.5-Math-1.5B** model. We also extended our evaluation to the larger **Qwen2.5-Math-7B model** (Yang et al., 2024). Since the default maximum sequence length of Qwen2.5-Math-7B (4096) was insufficient for our tasks, we expanded its RoPE theta from 10,000 to 40,000 and increased the context window to 16,384, mirroring the setup for the 1.5B model.

For all RL-based training, we used a batch size of 16 and a fixed learning rate of $5 \times 10^{-6}$. All experiments were conducted on a setup with 2 × A800 GPUs. RL-based models were trained for 500 steps with 4 rollouts per prompt,while SFT models were trained for 3 epochs. Our implementation is based on the TRL library (Team, 2025), which utilizes vLLM (Kwon et al., 2023) as a generation server. We independently reproduced and validated the results for the Simpel-RL (Zeng et al., 2025), ReLIFT (Ma et al., 2025), LUFFY (Yan et al., 2025), and Chord-$\mu$ (Zhang et al., 2025b) baselines.

**LESE Hyperparameters:** The scaling factor $l$ was set to 1.8. Since the maximum possible total reward in our setting is 3.1, we set $D_{max}$ to the theoretical upper bound of the standard deviation, 1.55, to accommodate different reward function configurations and ensure that the normalized diversity metric maintains a reasonable dynamic range. Because only 4 samples were generated per prompt, the mastery signal was unstable, so we set $c = 0$. And we set $\epsilon_{low} = 0.8, \epsilon_{high} = 0.28$ following the settings in DAPO (Yu et al., 2025).

## A.3 EVALUATION OF 7B MODELS

Table 4: Performance comparison across different models and benchmarks.'–' indicates that the result was not reported in the corresponding paper.We report *avg@32* for AIME and AMC, and *pass@1* for the other benchmarks.

| Model | AIME24 | AIME25 | AMC23 | MATH-500 | GPQA | Avg. |
|---|---|---|---|---|---|---|
| Qwen-Math-7B | 11.4 | 7.0 | 52.1 | 68.4 | 24.7 | 32.7 |
| Qwen-Math-Instruct | 7.3 | 8.2 | 49.5 | 82.2 | 32.3 | 35.9 |
| SimpleRL-Zero | 17.6 | 9.8 | 59.9 | 76.8 | 27.8 | 38.4 |
| LUFFY | **18.6** | 14.1 | 64.8 | 84.2 | 38.9 | 44.1 |
| ReLIFT | 13.8 | 20.1 | 68.3 | 86 | 39.9 | 45.6 |
| Chord-$\mu$ | 18.1 | 17.9 | 60.8 | – | – | – |
| LESE (Ours) | 16.9 | **23.5** | **68.6** | **89.8** | **47.0** | **49.2** |

All evaluations were performed using the vLLM framework (Kwon et al., 2023) with a generation temperature of 0.6 and a maximum token limit of 8192. The correctness of final answers was programmatically verified using MATHVERIFY (Face, 2024). We have extended the method to the Qwen2.5-Math-7B model. Due to computational constraints, our results for the 7B baselines were obtained using publicly available checkpoints from Hugging Face; for models without such checkpoints, we cite the performance reported in their respective papers. As shown in Table 4, LESE demonstrates a more pronounced performance advantage at the 7B scale. Notably, LESE achieves a new state-of-the-art average score of 49.2, surpassing strong baselines like LUFFY (Yan et al., 2025) by 4.6 points and ReLIFT (Ma et al., 2025) by 3.1 points. It is crucial to highlight that this superior performance was achieved with greater computational efficiency. We reduced the number of rollouts per prompt from 8 (as used in the baseline methods' implementations) to 4, effectively halving the generation overhead during training while still delivering enhanced results.

## A.4 CHAT TEMPLATE

To encourage the model to generate structured reasoning and adhere to the desired output format, we employed a consistent chat template for all training and evaluation experiments, as shown below.

```
You are a helpful AI Assistant that provides detailed thought
processes and correct answers. Please respond in the following
format:
<think>
[Your step-by-step thinking process here]
</think>
<answer>
[Your clear solution and answer here, with the answer in \boxed{}]
</answer>
```

## A.5 REWARD FUNCTION

We designed a multi-component reward function. For the LUFFY (Yan et al., 2025), ReLIFT (Ma et al., 2025), and Chord-$\mu$ (Zhang et al., 2025b) baselines, we followed the reward settings from their respective papers. For our experiments, we used the sum of the following three components:

$$R_{\text{core}} = \begin{cases} 1, & \text{if the final answer is correct} \\ 0, & \text{otherwise} \end{cases} \tag{8}$$

$$R_{strict} = \begin{cases} 1, & \text{if the answer is correct and enclosed in the } <answer> \text{ tag} \\ 0, & \text{otherwise} \end{cases} \tag{9}$$

The $R_{format}$ setting is as follows:

- **Fine-grained Reward.** The model receives a partial reward of 0.1 for each of the following conditions being met: the presence of each required tag (`<think>,</think>,<answer>,</answer>`), the correct count (exactly one) for each tag, and the correct sequential order of the tags. This sums to a maximum of 1.1.
- **Binary Reward** The model receives a reward of 1.1 only if the entire output matches the regular expression `^<think>...\n</think><answer>...</answer>$` and all four tags appear exactly once.

## A.6 THE USE OF LARGE LANGUAGE MODELS (LLMS)

Throughout the entire research and writing cycle of this paper, Large Language Models (LLMs) were only used for the paper polishing process, and did not participate in any research ideation work.

