# OpenReview forum: "Learning from Examples and Self-Exploration: A New Paradigm for Dynamic Fusion"
_ICLR.cc/2026/Conference — ICLR 2026 Conference Withdrawn Submission_

### Official Review · Reviewer_6oJa · 2025-10-22

**Soundness:** 2
**Presentation:** 2
**Contribution:** 3
**Rating:** 2
**Confidence:** 4

**Summary:**

This paper proposes LESE (Learning from Examples and Self-Exploration), a way to train large language models by dynamically blending supervised fine-tuning and reinforcement learning. Instead of fixing their ratio (static), LESE adjusts it per example based on how well the model understands the task (task mastery) and how diverse its solutions are (exploration diversity). This helps the model stay stable when it’s uncertain and explore more once it’s confident. It also uses a filtering technique to filter out training on examples that already get good rewards. With fine-grained rewards for clearer feedback, LESE outperforms existing methods on several math benchmarks (ID, OOD), showing better reasoning, stability, and generalization.

**Strengths:**

I think this paper explores a really interesting direction by trying to **dynamically combine SFT and RLHF** instead of treating them as separate stages. The idea of adjusting their balance per instance based on the model’s current mastery and exploration diversity feels both intuitive and well-motivated. I also like the **filtering mechanism for saturated instances** — skipping updates when all responses already reach maximum reward is a clever and practical way to avoid overfitting and wasted computation, making the training process more efficient and stable.

**Weaknesses:**

1. The explanation of Exploration Diversity (D) is unclear. The equation introduces a summation over N without defining what N represents or how it relates to the reward components. It would help to clarify whether N is the number of reward signals or samples, and how exactly D is computed (per instance or across batches).

2. Figure 3 is confusing. What exactly does R_format represent? What are the y-axis units or limits? If I understand correctly, the GRPO curve actually rises faster at the beginning — so why does the text claim that “LESE converges rapidly and maintains stable preference rewards”? Do you mean a different phase of training or a different metric? It would help to clarify this or add a zoom-in or wall-clock-time plot to support the claim.

3. I’d like to see a concrete example comparing LESE with a standard SFT then RL pipeline on the same prompt. Showing the original model’s answer and how it improves through each training regime would make the adaptive mechanism much easier to understand.

4. Similarly, providing an example with outputs from SFT-only, RL-only, and LESE for one problem would help illustrate the differences in behavior — for instance, how SFT falls into the imitation trap, how RL might show exploration risks, and how LESE balances the two.

5. The experiments are limited to mathematical reasoning tasks. Have you tried LESE on other domains like coding or logical reasoning to test its generality? That would strengthen the paper’s broader claims.

6. It would be useful to quantify the rollout efficiency gains introduced by the “saturated instance filtering” mechanism. How many rollouts per prompt are actually skipped compared to standard SFT→RL training? Presenting this comparison would make the efficiency argument more convincing.

7. The Figure 4 caption seems incorrect for subfigures (c) and (f). Also, the color coding isn’t explained, making it hard to interpret. Please clarify which colors correspond to which methods, specify what metrics the axes represent, and add short explanations for each subfigure.

**Questions:**

Please take a look at weaknesses section. Also I think the overall writing of the paper needs to be improved a lot.

---

### Official Review · Reviewer_knXr · 2025-11-01

**Soundness:** 2
**Presentation:** 2
**Contribution:** 2
**Rating:** 2
**Confidence:** 4

**Summary:**

This paper introduces a method to adaptively combine SFT and RL - depending on mastery (M) - how often the model already solves the task and diversity (D) -  how much reward varies across samples, The authors propose a method to combine these metrics to set alpha as a param to interpolate between SFT Loss and RL Loss.
Intuitively, SFT is used when the model cannot solve the task naively, and RL, when the model can solve the task atleast once. Results are reported on math reasoning benchmarks.

**Strengths:**

The problem statement is well motivated, if solved could be impactful in how we can combine the strengths of imitation and on policy RL for language models.

**Weaknesses:**

1. Method generality
  * Most results are on Qwen2.5-Math-1.5B–style math solvers. While the results are positive, adding tasks such from reasoning gym, which are harder to solve would also improve confidence in the method. Qwen 2.5-1B is know to have poor initial performance on these tasks.

2.  Diversity collapse / pass@k vs pass@1
  * When $\alpha$ is low (ie. more SFT), do we see any loss of diversity in outputs?
  * Concretely: how does LESE affect pass@k (best-of-k sampling) vs pass@1? RL-style exploration often helps pass@k more than pass@1; heavy SFT pressure might over-regularize toward a single canonical answer style.
  * If entropy collapse is an issue with standard GRPO baseline, having the comparison would be helpful.

3. Choice of mastery threshold c, number of rollouts.
   * How is the mastery cutoff c picked in practice? Is it tuned per dataset, fixed globally, or annealed? Please report sensitivity to c, since $\alpha$ rises sharply once M crosses c. In the appendix, for the math specific task, c=0. This assumes that the model if unable to solve an instance will rely completely on SFT.
   * The appendix states that only 4 rollouts were used, can this be increased to mitigate the stability issues for RL mentioned in the Appendix? - An ablation on how number of rollouts affect the stability of the method would strengthen the paper.

4. Section 4.2 – using output length as confidence.
  * The authors claim that LESE gives short answers when confident and longer chains when uncertain. That’s plausible, but length is only a proxy for hardness/uncertainty. Stronger metrics would make the claim better.

5. Line 266: fine-grained reward
   * The authors mention a fine-grained / multi-component reward. Can you spell out the components (format correctness, reasoning validity, final answer match, etc.), how they’re combined, and whether weights are hand-tuned or learned? This is important for reproducibility and for understanding how generalizable the approach is beyond math-style verifiable tasks.
   * Details in appendix are for math related datasets, would similar reward functions have to be handcrafted for any RLVR task?

6. Out-of-domain alignment degradation.
  * To argue the SFT component isn’t overbearing, can you add evals on unrelated chat / instruction benchmarks (e.g. open-ended dialogue like WildChat, safety / helpfulness like MT-Bench) to check that LESE hasn’t harmed general usability? This would also help position the method as “alignment-friendly,” not just math tuning. Especially since high levels of SFT have been shown to hurt model performance.

7. Curriculum details (Appendix A).: The authors  mention a hand-crafted curriculum.
  * How exactly is that curriculum in the training data curated (sampling weights? staged unfreezing? ordering of problems)? Is it model specific? This is unclear from the details in the appendix
  * The related work section would benefit from a discussion on curriculum learning as well, and not just RLHF vs SFT.

**Questions:**

Please see weaknesses section.

---

### Official Review · Reviewer_k4Vi · 2025-11-01

**Soundness:** 2
**Presentation:** 2
**Contribution:** 2
**Rating:** 2
**Confidence:** 4

**Summary:**

This work proposes instance-level dynamic fusion of SFT (using expert examples) and RL (via self-exploration).
This is done by adaptively weighting their losses, based on task mastery and exploration diversity:
intuitively, the method increases SFT weight when task mastery is low or exploration diversity is limited, and increases RL weight otherwise.
Empirical comparison with prior SFT-RL fusion methods are conducted on math and reasoning benchmarks.

**Strengths:**

- Presents a reasonable motivation and practical approach for instance-level dynamic fusion of SFT and RL, in contrast to prior fusion methods that are largely static.
- Shows promising empirical performance compared to selected baselines.

**Weaknesses:**

**Concerns about experimental setup:**

- Different reward designs are used for baseline methods, according to Appendix A.5. This is problematic for fair comparison, as reward design is orthogonal to SFT–RL fusion strategy.
- No details are provided on how hyperparameters for baseline methods are selected or tuned.
- Experiments are conducted only with Qwen2.5-math-1.5B/7B models, and primarily on mathematics benchmarks (with the exception of GPQA), raising the question about how generalizable the results could be.
- In Section 4.1, the training set contains 32,000 samples, with an RL batch size of 16. This implies one epoch equals 2,000 steps, yet RL methods are trained for only 500 steps. The rationale for this choice is unclear, and it is uncertain whether the algorithms had converged before the reported results in Table 1 were obtained.
- Training samples are ordered according to an easy-to-hard curriculum, which appears non-standard and somewhat arbitrary.
- The number of rollout generations per sample is not reported.
- For the 7B model results (Line 364), the paper compares with baseline numbers from prior works. This is unlikely to be a fair or meaningful comparison, as differences in training data, curricula, and initial models (e.g., instruct vs. math models) introduce significant variability.



**Writing and presentation issues:**

- Lines 40–50: The discussion on limitations of SFT and RLVR (e.g., “imitation trap,” “exploration risks,” “vanishing gradients”) lacks citations.
- Lines 77 and 80 are repetitive.
- Line 116: The sentence listing PPO, DPO, and GRPO is grammatically awkward and unclear.
- Eq. (5): The variable $R_{j,i}$ is not clearly defined, though it seems to refer to the $i$-th sub-reward for the $j$-th response. Similarly, the meanings of $R_{\text{score}}, R_1, R_2$ in Figure 2 are unclear.
- Line 223: The claim that “the marginal utility of exploration grows non-linearly with task competence” does not seem intuitive. The proposed function in Eq. (6) seems more like an ad-hoc choice that grows with both $M$ and $D$.
- Line 278: The claim regarding GRPO’s failure with binary rewards is not supported by Figure 3.
- Line 349: The assertion that the method “adheres to stringent human preference rewards” is not supported by results in Table 1.
- Lines 357–358: The reported “4.5 and 2.9 points improvement” over baselines should clarify how these numbers are calculated exactly.
- Figure 4 caption:
  - (a, d): The test datasets are unspecified.
  - “(b, e) pass@k performance on the AIME benchmark” should read (c, f), and it is unclear whether AIME24 or AIME25 was used.
  - The 3D plots in (c, f) offer no apparent benefit over simpler 2D line plots.
- Line 455: The statement that LESE-Static and LESE-Random have “formatting optimized at the expense of reasoning” conflicts with their 0 format score in Table 3.



**Other concerns:**


- The proposed method introduces multiple new hyperparameters (e.g., $R_{\text{thres}}$ in Line 192; $c$ and $\ell$ in Line 215) with limited guidance on their selection.
- The second claimed contribution (“fine-grained preference reward design”) is not novel, as reward design has long been a standard RL practice.
- The notion of “human preference” in this paper is vague; based on Line 269, it appears to mainly refer to formatting constraints (e.g., tag presence, ordering, format compliance).
- The paper does not discuss its limitations, despite clear issues raised above.

**Questions:**

Please see "weaknesses" above.

---

### Official Review · Reviewer_N8nZ · 2025-11-03

**Soundness:** 1
**Presentation:** 2
**Contribution:** 2
**Rating:** 2
**Confidence:** 3

**Summary:**

This paper proposes to combine the RL and SFT losses as a convex combination for training LLMs on reasoning datasets. The goal of this combination is to compensate for the weaknesses of individual methods: SFT alone leads to overfitting, while RL alone leads to reward hacking or sample inefficiency. The convex combination parameter, $\alpha$, is composed of two components: task mastery and exploration diversity, which are dynamically learned during the training of the LLM. This approach, when paired with “fine-grained rewards” (a concept the paper references but does not detail), resulted in improved performance on reasoning benchmarks under a specific training setup.

**Strengths:**

- The core idea of combining the RL and SFT losses to harness the strengths of both methods is a promising approach, as it allows both successful and unsuccessful trajectories to provide useful gradient updates to the LLM.
- The proposed approach dynamically combines these losses (rather than using a static combination), which reportedly results in faster training by weighting one loss over another based on question difficulty and the model's competence.
- The results demonstrate improved performance under a specific training setup over its baselines on a variety of math reasoning benchmarks, including AIME and MATH-500.

**Weaknesses:**

- The approach appears mathematically unsound. There is more nuance to SFT and RL-based training approaches that requires careful and thorough investigation (e.g., [1]). Trivially combining the losses is methodologically questionable, as the units of the two losses are incommensurable (analogous to adding kilograms and kilometres).
- The paper lacks a clear explanation or mathematical insight for Eq. 6. Furthermore, it does not justify why the proposed estimation method for the $\alpha$ parameter is appropriate or correct.
- The experiments are performed in a non-standard setup (lines 306-312), and the results, including those of baselines, differ drastically from the standard, publicly reported results on Huggingface for the same 1.5B model (https://huggingface.co/deepseek-ai/DeepSeek-R1-Distill-Qwen-1.5B).
- The presentation in some parts of the draft is unprofessional. For example, the plots lack titles and y-axis labels, and appear to be unprocessed screenshots. There are no discussion or limitation sections in the paper.

[1] Roux, Nicolas Le, et al. "Tapered off-policy reinforce: Stable and efficient reinforcement learning for LLMs." arXiv preprint arXiv:2503.14286 (2025).

**Questions:**

- The discussion of token lengths in Table 2 presents a counterintuitive finding. Intuitively, RL approaches should have higher token counts due to exploration, while SFT methods should have fewer because the LLM imitates a known answer. The table, however, shows the opposite. The authors should explain this discrepancy. Furthermore, the relevance of this discussion on token length is unclear, given that the proposed method does not explicitly optimize for it.
- The authors should address the apparent poor performance of the standalone RL method in Figures c and f (compared to SFT and LESE). This result is counterintuitive, given that RL methods are generally expected to generalize better and eventually outperform SFT-only approaches.
- The paper requires a detailed explanation of the “fine-grained rewards” referenced in the text. Specifically, the authors should describe: (1) what these multiple reward signals are, (2) how they are combined with the true reward, and (3) what mechanisms are in place to prevent reward hacking, especially given their heuristic nature.

---

### Note · Authors · 2026-01-04

I have read and agree with the venue's withdrawal policy on behalf of myself and my co-authors.